# Association of prior opium addiction with incident non-alcoholic fatty liver disease: A case-control study

Sina Bazmi[1,2], Talieh Parvaneh[2,3], Zahra Mohammadi[2], Parisa Pishdad[4]*,
Mehrnaz Ghaedian[5,6], Azizallah Dehghan[7], Reza Homayounfar[8], Mojtaba Farjam[7,3]*

1 USERN Office, Fasa University of Medical Sciences, Fasa, Iran, 2 Student Research Committee, Fasa University of Medical Sciences, Fasa, Iran, 3 Clinical Research Development Unit, Valiasr Hospital, Fasa University of Medical Sciences, Fasa, Iran, 4 Medical Imaging Research Center, Department of Radiology, Shiraz University of Medical Sciences, Shiraz, Iran, 5 Department of Radiology, School of Medicine, Fasa University of Medical Sciences, Fasa, Iran, 6 Department of Radiology, Namazi Hospital, Shiraz University of Medical Sciences, Shiraz, Iran, 7 Noncommunicable Diseases Research Center, Fasa University of Medical Science, Fasa, Iran, 8 National Nutrition and Food Technology Research Institute (WHO Collaborating Center), Faculty of Nutrition Sciences and Food Technology, Shahid Beheshti University of Medical Sciences, Tehran, Iran

* Farjam.phd@gmail.com (MF); p_pishdad@yahoo.com (PP)

## Abstract

### Background

Non-alcoholic fatty liver disease (NAFLD) is a prevalent chronic liver condition with no approved pharmacological treatments. Given opium's potential metabolic effects on lipid profiles, blood pressure, and glucose levels, factors known to influence NAFLD, we hypothesized that opium addiction might be inversely associated with NAFLD risk.

### Objective

To investigate the association between opium addiction during a six-year period (2016–2022) and the subsequent incidence and severity of NAFLD in 2022 among participants of the Fasa Adult Cohort Study (FACS).

### Methods

Adults aged 35–70 were selected from the FACS baseline dataset (2016) after excluding individuals with NAFLD (based on the Fatty Liver Index and regional cut-offs), obesity, cancer, chronic liver diseases, or regular alcohol use. Of 550 randomly selected participants invited for sonography in 2022, 396 attended; 170 were newly diagnosed with NAFLD. Cases and controls were matched 1:1 using SPSS based on age, sex, diabetes, and hyperlipidemia. Opium addiction was defined using DSM-5 criteria via structured interviews, while NAFLD diagnosis and grading were performed

**Data availability statement:** Due to institutional policy, the dataset generated and analyzed during the current study is not publicly available. Data access is restricted and governed by a data and material transfer agreement that prohibits further transfer without prior written consent from the data provider. However, the data can be made available upon reasonable request to the Fasa Non-Communicable Diseases Research Center (NCDRC) Management Team, which serves as the institutional contact point. Requests may be directed via telephone at +98 71 5331 4068 or via email at ncdrc.fums.ac.ir@gmail.com. Please cite this article when making your request.

**Funding:** The author(s) received no specific funding for this work.

**Competing interests:** The authors have declared that no competing interests exist.

using blinded ultrasound assessment. Fisher's exact and Fisher-Freeman-Halton tests were used for analysis. A post hoc power analysis was also conducted.

## Results

The final analysis included 206 participants (103 cases, 103 controls). Opium addiction was observed in 31 NAFLD cases and 72 controls, a non-significant difference. However, the prevalence of opium addiction differed significantly across NAFLD severity grades. The post hoc statistical power was estimated at 60%.

## Conclusion

Although not statistically significant, fewer opium addicts developed NAFLD than non-addicts. This inverse trend, along with significant variation across NAFLD grades, suggests a possible association that warrants further investigation. Larger studies are needed to explore this potential relationship. If confirmed, opioid-based therapies may offer dual benefits for managing chronic pain and metabolic risk in selected NAFLD populations.

## Introduction

Non-alcoholic fatty liver disease (NAFLD) is the most prevalent chronic disease of liver globally [1] characterized by excessive hepatic fat accumulation in the absence of significant alcohol consumption or other secondary causes of hepatic steatosis, with an escalating incidence rate. It is estimated that currently, 32% of adults worldwide are affected by NAFLD [2]. NAFLD encompasses a spectrum of liver pathology ranging from simple steatosis to non-alcoholic steatohepatitis (NASH) [3], which may progress to liver fibrosis, cirrhosis (a deadly condition), or liver cancer [4]. In addition to liver-related morbidity and mortality, it has been demonstrated to elevate the risk of cardiovascular diseases [5]. Due to its largely asymptomatic nature in early stages and its association with preventable metabolic risk factors [6], NAFLD is increasingly recognized as a critical target for public health interventions and policy planning. Early identification and modification of modifiable risk factors are vital to mitigating its burden. Moreover, despite significant advancements in understanding the mechanisms involved in NAFLD pathogenesis and progression, there is currently no approved pharmacological treatment for it [7]. The management goal focuses on multidimensional weight reduction and the reduction of cardiometabolic risk factors such as blood lipid levels, blood pressure, and blood glucose [8].

The major opium market is located in Asia, and in various Asian countries, there has been an ancient belief that opium is a preventive and curative agent for many diseases [9]. Essentially, the name "Theriac" has been chosen as its old Persian name, which means antidote. Today, some Asians, including Iranians, still believe that opium can attenuate serum lipids, blood pressure, and blood glucose [9]. Given the social harm and widespread market of opium, numerous studies have been

conducted to examine the validity of this belief, yielding contradictory results [9]. Some studies still report findings supporting its positive effects on these factors [10].

Considering the strong association of elevated serum lipids, blood sugar, and blood pressure with NAFLD [11–13], we hypothesized for the first time that opium may be linked to a reduced risk of NAFLD. We conducted this investigation in a rural population from Fasa. Previously, a reduction in serum lipids was reported in opium users in the same community [10].

Despite the growing burden of NAFLD and its intersection with metabolic health, there is limited evidence on the potential role of unconventional exposures, such as opium addiction, in its pathogenesis. Opium use remains a culturally entrenched practice in parts of the Middle East and South Asia, including Iran, where its medical and recreational consumption is historically widespread. While previous research has examined opium in relation to cardiovascular, metabolic, and neoplastic outcomes, its longitudinal association with newly developed NAFLD has not been explored. To the best of our knowledge, this is the first study to evaluate the relationship between opium addiction and incident NAFLD diagnosed via sonography in a population-based setting. By using a rigorously selected, matched case-control design within a well-characterized cohort, our study seeks to fill a critical gap in the literature and inform future longitudinal investigations on the metabolic consequences of opiate use.

## Materials and methods

This study is a matched case-control survey aimed at investigating the link between opium addiction in recent years and the incidence and severity of NAFLD.

### Study population and design

Participants were selected from the Fasa Adult Cohort Study (FACS), a population-based cohort of individuals aged 35–70 residing in the rural area of Sheshdeh and its 24 surrounding villages. The initial phase of FACS was conducted in 2015–2016, during which detailed demographic, clinical, and laboratory data were collected. All participants provided written informed consent, and study protocols were documented and registered [14].

From the 2016 FACS dataset, individuals with suspected NAFLD were excluded using the Fatty Liver Index (FLI) and sex-specific, geographically validated cutoff points [15] As sonographic evaluation was not performed in 2016, FLI was used to reasonably exclude prevalent NAFLD cases. Further exclusion criteria included: body mass index (BMI) >30 kg/m², diagnosis of chronic liver disease (e.g., hepatitis B or C), any form of malignancy, or regular alcohol consumption, to eliminate major confounding factors for hepatic steatosis. From the eligible population, 550 individuals were randomly selected via computer-generated sampling and invited to Vali-Asr Hospital in Fasa for follow-up evaluations between July 5 and September 6, 2022. A total of 396 individuals participated in this follow-up, where liver ultrasonography was performed to identify incident cases of NAFLD. Among these, 170 individuals were newly diagnosed with NAFLD based on ultrasound. A control group of 220 individuals without NAFLD was identified. To minimize confounding, 103 case participants were matched 1:1 with 103 control participants based on key variables influencing NAFLD risk, age, sex, diabetes status, and hyperlipidemia, using SPSS software's case-control matching function.

### Covariates

Baseline characteristics, including marital status, educational attainment, smoking status, daily caloric intake (kilocalories per 24 hours), history of diabetes, and lipid profiles, were obtained from the 2016 FACS dataset. Hyperlipidemia was classified as present if any of the following criteria were satisfied: high-density lipoprotein ≤ 40 mg/dL for men or ≤ 50 mg/dL for women, total cholesterol ≥ 200 mg/dL, triglycerides ≥ 150 mg/dL, low-density lipoprotein ≥ 130 mg/dL, or reporting the consumption of lipid-lowering medication [16]. Diabetes was defined by physician diagnosis, fasting blood glucose ≥126 mg/dL, or the use of insulin or oral hypoglycemic agents. Daily calorie intake was estimated using a validated

semi-quantitative 125-item food frequency questionnaire (FFQ) developed for the Iranian population, which assessed average consumption of a wide range of food items over the past year. The FFQ responses were converted into energy intake (kilocalories per day) using standard portion sizes and the Iranian Food Composition Table.

## Exposure assessment

Opium addiction was assessed in 2022 by trained interviewers using DSM-5 criteria for opioid use disorder, which includes 11 symptoms spanning physiological, cognitive, and social domains [17]. A diagnosis of addiction required the presence of at least two criteria over a 12-month period between 2016 and 2022. The interviews were conducted in a structured format with standardized checklists.

## Outcome assessment

NAFLD was diagnosed in 2022 via transabdominal ultrasound, the preferred first-line modality for hepatic steatosis in clinical and epidemiologic settings. Diagnosis required sonographic evidence of hepatic steatosis (>5–10% liver fat content) in the absence of other liver diseases or significant alcohol intake (<30 g/day for men and <20 g/day for women), both of which had already been excluded [18]. Since ultrasound is utilized as the primary imaging method in patients with suspected NAFLD [19], it was chosen due to its accessibility and cost-effectiveness for rural populations. The diagnosis of NAFLD was validated through ultrasound findings indicating hepatic steatosis >5–10% of the liver's weight [20]. All sonographic assessments were performed by a single radiologist blinded to participants' exposure status. NAFLD severity was further graded using the Matteoni classification, which stratifies disease into four histopathologically informed classes: Class 1: Simple steatosis without inflammation and fibrosis, Class 2: Steatosis with lobular inflammation but without fibrosis, Class 3: Presence of ballooned hepatocytes, and Class 4: Presence of either Mallory bodies or fibrosis [21].

## Statistical analyses

Qualitative information was depicted through numerical values and percentages, whereas quantitative data was illustrated using average values and standard deviations. Fisher's exact test was employed to assess the numerical and percentage differences of qualitative factors between the case and control groups, while the independent t-test was utilized to compare the average values of quantitative factors.

Using the case-control matching feature in SPSS software, the case group (individuals with NAFLD) and the control group (individuals without NAFLD) were matched based on age, gender, diabetes, and hyperlipidemia (Table 1). This process reduced the sample size to 206 individuals (103 in the case category and 103 in the control category).

We used the Fisher's exact test to compare the numbers and percentages of opium-addicted individuals between the matched case and control groups. Additionally, we used Fisher-Freeman-Halton exact test to compare the numbers and percentages of opium-addicted individuals among different grades of NAFLD. We chose this approach to emphasize direct group comparisons after strict matching, rather than multivariable regression modeling, in order to preserve transparency in the effect of opium exposure between groups. Although regression models could be applied, our design and analytical strategy were tailored to highlight differences in exposure prevalence in a matched observational framework.

All analyses were carried out using SPSS v 23, with significance level for all tests set as 0.05. Furthermore, at the end of the study, power analysis was conducted using the statsmodels library in the Python programming language to ensure an adequate sample size for achieving sufficient statistical power.

## Ethics approval and consent to participate

The research protocol of this study was approved by both the Research Council and Ethics Committee of Fasa University of Medical Sciences, with the approval code and grant number of IR.FUMS.REC.1401.022 and IR.FUMS.REC.1400.027.

 

**Table 1. Case and control group baseline characteristics.**

| Variable | | Cases (NAFLD +)[a] | Controls (NAFLD -)[a] | Total | P-value[b] |
|---|---|---|---|---|---|
| **Gender** | Male | 46 (22.3%) | 46 (22.3%) | 92 (44.7%) | 1.00 |
| | Female | 57 (27.7%) | 57 (27.7%) | 114 (55.3%) | |
| **Age (years)** | | 47.80 ± 7.99 | 47.80 ± 7.99 | 47.80 ± 7.97 | 1.00 |
| **Diabetes** | Yes | 10 (4.9%) | 10 (4.9%) | 20 (9.7%) | 1.00 |
| | No | 93 (45.1%) | 93 (45.1%) | 186 (90.3%) | |
| **Hyperlipidemia** | Yes | 86 (41.7%) | 86 (41.7%) | 172 (83.5%) | 1.00 |
| | No | 17 (8.3%) | 17 (8.3%) | 34 (16.5%) | |
| **Smoker** | Yes | 33 (16%) | 26 (12.6%) | 59 (28.6%) | 0.355 |
| | No | 70 (34%) | 77 (37.4%) | 147 (71.4%) | |
| **Highest education** | Elementary school | 41 (19.9%) | 37 (18%) | 78 (37.9%) | 0.421 |
| | Middle school | 14 (6.8%) | 11 (5.3%) | 25 (12.1%) | |
| | High school diploma | 3 (1.5%) | 8 (3.9%) | 11 (5.3%) | |
| | Illiterate | 45 (21.8%) | 47 (22.8%) | 92 (44.7%) | |
| **Marital status** | Single (never married) | 1 (0.5%) | 2 (1%) | 3 (1.5%) | 0.761 |
| | Married | 100 (48.5%) | 98 (47.6%) | 198 (96.1%) | |
| | Widowed | 2 (1%) | 3 (1.5%) | 5 (2.4%) | |
| **Daily calorie intake (kcal)** | | 2771.20 ± 1025.17 | 2773.81 ± 1108.00 | 2772.51 ± 1064.79 | 0.986 |

[a]Case and Control groups matched for age, gender, diabetes, and hyperlipidemia.

Abbreviations: NAFLD: Non-alcoholic fatty liver disease.

NAFLD is defined by sonography imaging.

[b]Fisher's Exact Test for categorical variables and independent-samples t test for the continuous variable, significance level set at 0.05.

The study followed the principles set forth in the Helsinki Declaration. Before taking part in the study in 2022, all participants were briefed on the research goals and provided written informed consent. The authors do not have access to any information that might identify individual participants following data collection.

## Results

Eventually, 206 individuals (mean age 47.80 ± 7.99) were included in the analyses, with 103 individuals in the case group with NAFLD and 103 individuals in the control group. The study consisted of 92 males (44.7%) and 114 females (55.3%), with 46 males and 57 females in both case and control groups. In both the case and control groups, there were 10 diabetic patients and 86 hyperlipidemia patients. Furthermore, no notable distinctions existed between the groups concerning the quantity of individuals who smoke, educational level, marital status, and daily caloric intake of the diet (Table 1).

Seventy individuals (34% of the study population) were addicted to opium. Among them, 31 individuals (15%) in the case group (NAFLD group) and 72 individuals (35%) in the control group were opium addicts. Therefore, fewer opium addicts were affected by NAFLD, while more non-opium users were affected by NAFLD (Fig 1). However, this difference was not statistically significant (Table 2).

After determining the grading of NAFLD, 103 individuals were classified as normal, while 79 individuals (38.3%) had grade 1, and 24 individuals (11.6%) had grades 2 and 3. No considerable differences were found in any of the study variables between the groups (Table 3). However, there was a difference in the number of opium addicts among different grades of NAFLD (Fig 2), which was statistically significant (Table 4).

To ensure an adequate sample size for achieving sufficient statistical power, a power analysis was conducted. Assuming a power of 0.80, a significance level of 0.05, and expecting a moderate effect size (Cramer's V = 0.3), each group

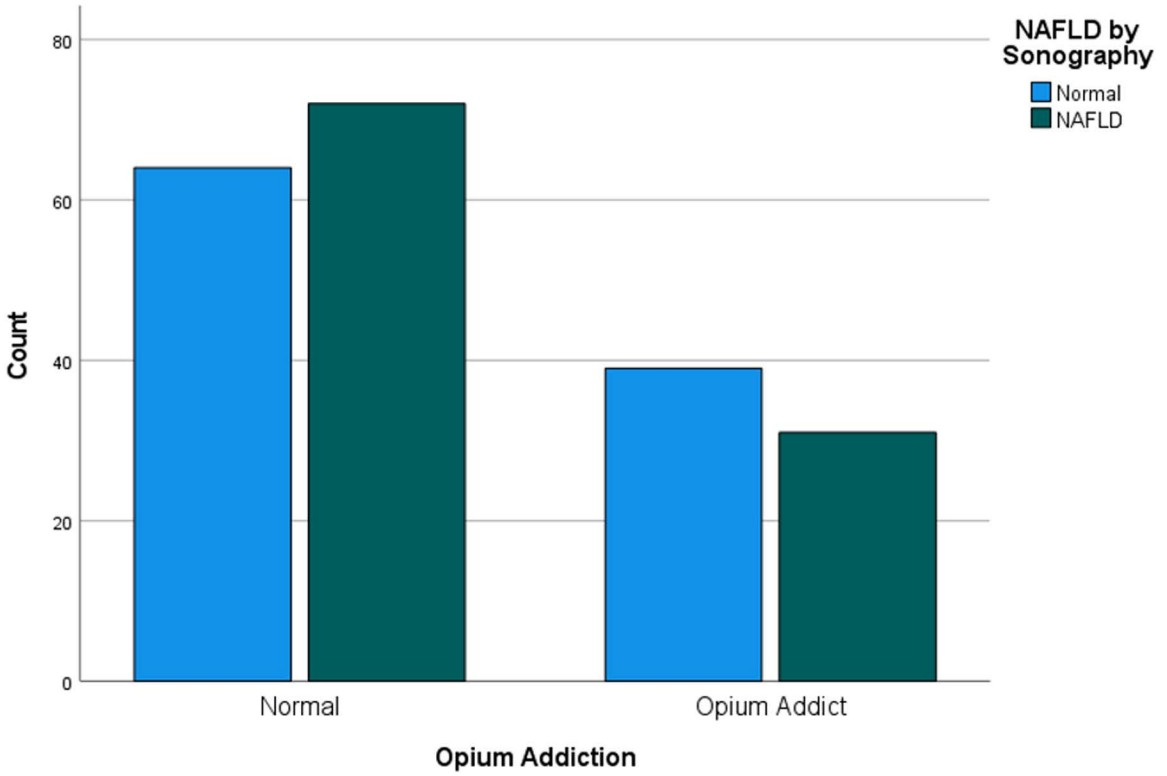

**Fig 1. NAFLD incidence in opium addicts and non-addicts.**

**Table 2. Opium addiction among the case and control groups.**

| Variable | | Cases (NAFLD +)[a] | Controls (NAFLD -)[a] | Total | P-value[b] |
|---|---|---|---|---|---|
| Opium Addiction | Yes | 31 (15%) | 39 (18.9%) | 70 (34%) | 0.303 |
| | No | 72 (35%) | 64 (31.1%) | 136 (66%) | |
| Total | | 103 (50%) | 103 (50%) | 206 | |

[a]Case and Control groups matched for age, gender, diabetes, and hyperlipidemia.

Abbreviations: NAFLD: Non-alcoholic fatty liver disease.

NAFLD is defined by sonography imaging.

[b]Fisher's Exact Test, significance set at 0.05.

required a minimum of 122 individuals. Based on this analysis, a sample size of 103 individuals in each group does not appear to be sufficient, resulting in a statistical power of 60%, which falls below the desired power of 80%.

## Discussion

This case-control study sought to examine the connection between opium addiction in recent years and newly diagnosed NAFLD. After excluding individuals with chronic liver diseases, alcohol consumption, and obesity, the remaining participants were split into two groups: those with NAFLD and those without NAFLD. The most influential factors on NAFLD, including gender, age, hyperlipidemia, and diabetes, were matched using software. Additionally, it was ensured that there were no considerable differences in other background variables between the two groups. The results of this study showed

 

**Table 3. Study population baseline characteristics with regards to NAFLD grades.**

| Variable | | NAFLD considering grades[a] | | | Total | P-value[b] |
|---|---|---|---|---|---|---|
| | | Normal | Grade 1 | Grade 2–3 | | |
| Gender | Male | 46 (22.3%) | 37 (18%) | 9 (4.4%) | 92 (44.7%) | 0.734 |
| | Female | 57 (27.7%) | 42 (20.4%) | 15 (7.3%) | 114 (55.3%) | |
| Age (years) | | 47.80 ± 7.99 | 48.20 ± 7.68 | 46.46 ± 8.97 | 47.80 ± 7.97 | 0.646 |
| Diabetes | Yes | 10 (4.9%) | 6 (2.9%) | 4 (1.9%) | 20 (9.7%) | 0.394 |
| | No | 93 (45.1%) | 73 (35.4%) | 20 (9.7%) | 186 (90.3%) | |
| Hyperlipidemia | Yes | 86 (41.7%) | 64 (31.1%) | 22 (10.7%) | 172 (83.5%) | 0.535 |
| | No | 17 (16.5%) | 15 (7.3%) | 2 (1%) | 34 (16.5%) | |
| Smoker | Yes | 26 (12.6%) | 27 (13.1%) | 6 (2.9%) | 59 (28.6%) | 0.417 |
| | No | 77 (37.4%) | 52 (25.2%) | 18 (8.7%) | 147 (71.4%) | |
| Highest education | Elementary school | 37 (18%) | 31 (15%) | 10 (4.9%) | 78 (37.9%) | 0.621 |
| | Middle school | 11 (5.3%) | 9 (4.4%) | 5 (2.4%) | 25 (12.1%) | |
| | High school diploma | 8 (3.9%) | 3 (1.5%) | 0 (0%) | 11 (5.3%) | |
| | Illiterate | 47 (22.8%) | 36 (17.5%) | 9 (4.4%) | 92 (44.7%) | |
| Marital status | Single (never married) | 2 (1%) | 1 (0.5%) | 0 (0%) | | 1.00 |
| | Married | 98 (47.6%) | 76 (36.9%) | 24 (11.7%) | 198 (96.1%) | |
| | Widowed | 3 (1.5%) | 2 (1%) | 0 (0%) | 5 (2.4%) | |
| Daily calorie intake (kcal) | | 2773.81 ± 1108.00 | 2786.90 ± 1044.83 | 2719.54 ± 977.25 | 2772.51 ± 1064.79 | 0.964 |

[a]NAFLD grading is defined by sonography imaging.

Abbreviations: NAFLD: Non-alcoholic fatty liver disease.

[b]Fisher-Freeman-Halton Exact test for categorical variables and one-way ANOVA test for the continuous variable, significance level set at 0.05.

that a smaller number of individuals addicted to opium developed NAFLD, while a larger number of non-opium addicts were diagnosed with NAFLD. However, this association was not statistically significant. Due to the limited sample size and insufficient power of the study, it is possible that even if a significant relationship exists, our study was unable to detect it. Therefore, investigations with larger sample sizes should be performed. Furthermore, a significant difference was noted in the number of opium addicts among different grades of NAFLD, but the direction of this difference is not clear. Nonetheless, this finding can serve as a hypothesis for further studies.

No studies have evaluated the link between opium use and NAFLD to date. Animal studies have reported a significant increase in liver enzymes due to chronic opiate use in rats [22], while no significant difference in liver damage was found between cigarette and opium use in golden hamsters [23]. Pawan et al.'s study on 25 opium-addicted men showed a significant increase in liver enzymes compared to the control group [24]. However, the small sample size and single-gender population of this study reduce the reliability of its results, and elevated liver enzymes do not necessarily indicate a diagnosis of NAFLD. In Agbalajob et al.'s study [25], which was conducted on individuals with chronic liver diseases, despite a direct association between opium use and liver cancer and cirrhosis, a reverse relationship was found with NAFLD. This analysis was based on a one-year retrospective assessment, considering NAFLD as exposure and opium consumption as an outcome, and it cannot be used as direct evidence for the opposite direction of this relationship. However, further investigation is needed to delve into the possible causal and reciprocal connection between opium consumption and NAFLD, as it is possible that opium users may be less prone to NAFLD. Additionally, Nakhostin-Ansari et al.'s study [26] on the general population calculated the FIB-4 score, which uses platelet count, ALT, AST, and age as a measure of liver fibrosis risk, and opium use history was associated with higher FIB-4 scores. However, this analysis is cross-sectional, and individuals with higher FIB-4 scores, likely indicating more advanced chronic liver diseases, may have had higher

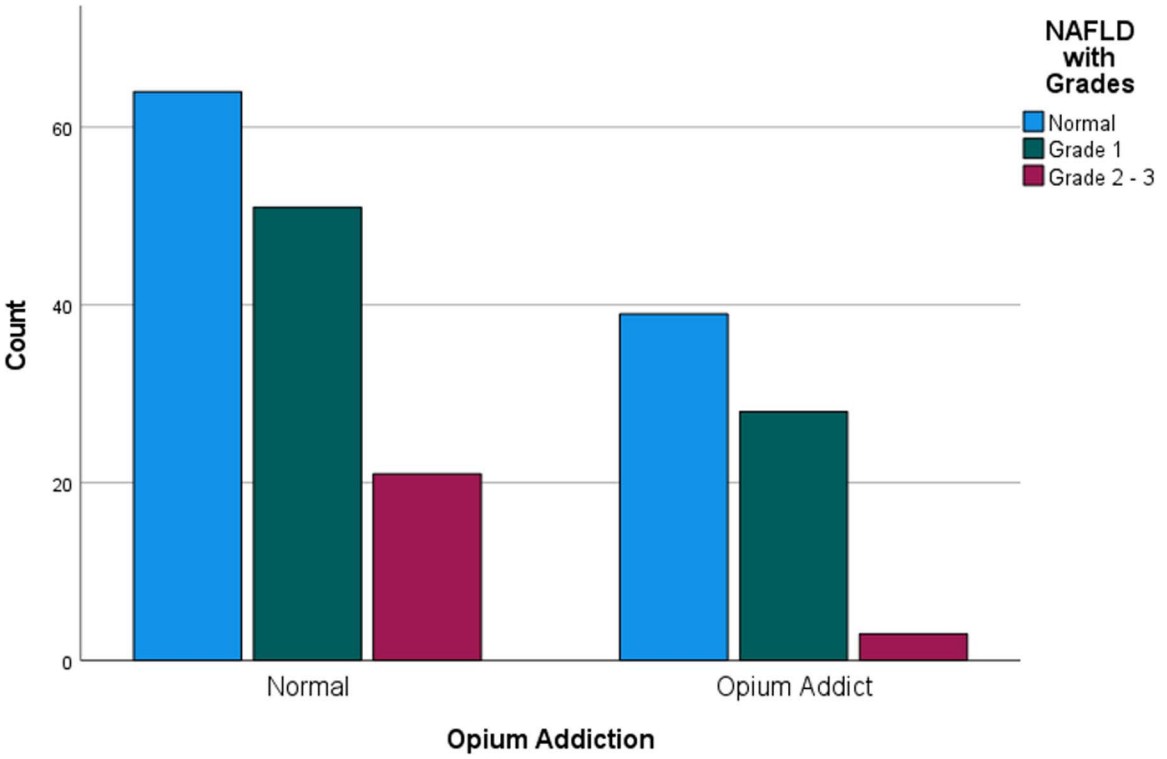

**Fig 2. NAFLD gradings among opium addicts and non-addicts.**

**Table 4. Opium addiction among NAFLD grades.**

| Variable | | NAFLD considering grades[a] | | | Total | P-value[b] |
|---|---|---|---|---|---|---|
| | | Normal | Grade 1 | Grade 2–3 | | |
| **Opium Addiction** | Yes | 39 (18.9%) | 28 (35.4%) | 3 (1.5%) | 70 (34%) | **0.046** |
| | No | 64 (31.1%) | 51 (24.8%) | 21 (10.2%) | 136 (66%) | |
| Total | | 103 (50%) | 79 (38.3%) | 24 (11.7%) | 206 | |

[a]NAFLD grading is defined by sonography imaging.

Abbreviations: NAFLD: Non-alcoholic fatty liver disease.

[b]Fisher-Freeman-Halton Exact Test, significance set at 0.05.

opium consumption due to chronic pain. Moreover, FIB-4 is solely based on laboratory findings and is not as reliable as direct imaging with sonography.

Two related cross-sectional studies have been conducted on the FACS-covered population. Kazemi et al.'s study [10] found that opium use declined the blood lipid levels, and Bijani's study [27] indicated a decrease in liver enzymes associated with opium use. These studies raised the hypothesis of a protective association between opium use and NAFLD, which is characterized by increased liver enzymes and elevated blood lipid levels in this population. This prompted us to investigate this association in a more valid manner, such as assessing NAFLD after opium exposure. Furthermore, the findings of the studies indicate that liver enzymes and other laboratory tests may yield false abnormalities in opium users. Therefore, apart from direct imaging, no other diagnostic tool can be relied upon for diagnosing NAFLD in opium consumers. Although we had the necessary parameters to calculate the FLI and define NAFLD based on it in the large

FACS population, we chose to use only sonography for diagnosing NAFLD in our study to avoid unreliable results, despite limitations.

Although opioids and opiates are typically recognized for their analgesic properties and lack of anti-inflammatory characteristics, studies have revealed that they may possess anti-inflammatory properties [28]. In fact, there is potential for their development into immunomodulatory medications for treating rheumatic diseases [29]. The involvement of inflammation in the development of NAFLD has been firmly established [30], suggesting that this potential could be explored for the management of NAFLD. Additionally, considering the importance of controlling blood lipids, blood pressure, and blood sugar levels in managing and preventing the progression and complications of NAFLD [8], opium holds the potential to address these factors.

In this study, we made significant efforts to conduct reliable analyses with a precise design. Additionally, we used Schlesselman's formula, which determines the sample size for case-control studies, to establish the minimum desired sample size. Unfortunately, an insufficient number of individuals participated in our study, resulting in a low sample size and reducing the study's statistical power. Consequently, even if a significant relationship exists, our study may fail to detect it. As shown in the bar chart, a protective association between opium addiction and the occurrence of NAFLD seems apparent. Therefore, it is recommended that subsequent research involve larger sample sizes to enhance the precision of their analyses to detect any significant associations more effectively. Furthermore, due to the significant relationship observed among NAFLD grades, we hypothesize that opium consumption may contribute to the progression of NAFLD. However, the accuracy and direction of this relationship are not clearly defined in our study, and it can only suggest a hypothesis for the design of future studies. Additionally, prolonged use of opium and its derivatives can potentially result in reduced physical activity, weight gain, and increased appetite [31]. These factors may contribute to hepatic lipid accumulation and could have influenced the results of our study, making it important to consider the duration of opioid use in future studies.

If a considerable link between opium use and the presence and severity of NAFLD is detected, considering that controlling blood lipid levels, blood pressure, and blood glucose is necessary for managing NAFLD and preventing its progression and complications [8], opium may potentially address all these aspects. Clinical trials can then investigate the long-term and short-term effects and safety of opioid-derived medications in managing risk factors and NAFLD progression in a subgroup of non-advanced NAFLD patients who experience chronic pain and have a lower risk of addiction. This research can contribute to the development of comprehensive pain management protocols and their integration into existing algorithms. However, if our objectives continue, opium and its derivatives should not be used as primary substances due to their well-known adverse social and health impacts, as studies have demonstrated the detrimental long-term effects of these substances on various organs, and even on the mentioned blood lipid levels, blood glucose, and blood pressure [9]. Opioid derivatives should be prescribed as approved pharmacological products under medical supervision [25]. Future studies can further explore the biochemical pathways influenced by opioids that may impact liver health.

In conclusion, opium may have a protective effect on NAFLD and its severity, but this association needs to be investigated through larger population studies and subsequent clinical trials using pharmacological opioid derivatives. This potential can be harnessed in the future to develop new algorithms for managing chronic pain in NAFLD patients or those at high risk for NAFLD but with a low risk of addiction. This study is the first to explore this association using incident NAFLD cases confirmed by sonography. Our findings underscore the need for larger, prospective studies to assess the metabolic impacts of opium use and help guide public health recommendations in regions where opiate consumption is prevalent.

## Acknowledgments

The authors would like to extend their gratitude to all individuals who participated in this study and generously provided honest and sincere responses to our inquiries. We also sincerely thank the Clinical Research Development Unit of Valiasr Hospital, Fasa University of Medical Sciences, for their valuable support in facilitating the research process.

## Author contributions

**Conceptualization:** Mehrnaz Ghaedian, Mojtaba Farjam.

**Data curation:** Talieh Parvaneh.

**Formal analysis:** Sina Bazmi.

**Investigation:** Sina Bazmi, Talieh Parvaneh, Zahra Mohammadi.

**Methodology:** Sina Bazmi.

**Project administration:** Mojtaba Farjam.

**Resources:** Mehrnaz Ghaedian, Azizallah Dehghan, Reza Homayounfar, Mojtaba Farjam.

**Software:** Sina Bazmi.

**Supervision:** Parisa Pishdad, Mehrnaz Ghaedian, Azizallah Dehghan, Reza Homayounfar, Mojtaba Farjam.

**Validation:** Sina Bazmi, Parisa Pishdad, Azizallah Dehghan, Reza Homayounfar, Mojtaba Farjam.

**Visualization:** Sina Bazmi.

**Writing – original draft:** Sina Bazmi, Zahra Mohammadi.

**Writing – review & editing:** Sina Bazmi, Parisa Pishdad, Azizallah Dehghan, Mojtaba Farjam.

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
