## [Decision Letter · Decision Letter 0]

Dear Dr. Farjam,

Thank you for submitting your manuscript to PLOS ONE. After careful consideration, we feel that it has merit but does not fully meet PLOS ONE’s publication criteria as it currently stands. Therefore, we invite you to submit a revised version of the manuscript that addresses the points raised during the review process.

We look forward to receiving your revised manuscript.

Kind regards,

Borhan Mansouri

Academic Editor

PLOS ONE

Journal Requirements:

Reviewers' comments:

Reviewer's Responses to Questions

**Comments to the Author**

1. Is the manuscript technically sound, and do the data support the conclusions?

Reviewer #1: Yes

Reviewer #2: Yes

2. Has the statistical analysis been performed appropriately and rigorously?

Reviewer #1: N/A

Reviewer #2: No

3. Have the authors made all data underlying the findings in their manuscript fully available?

Reviewer #1: Yes

Reviewer #2: Yes

4. Is the manuscript presented in an intelligible fashion and written in standard English?

Reviewer #1: No

Reviewer #2: Yes

Reviewer #1: The submitted manuscript (Submission Number: PONE-D-24-50823) describes the impact of opium consumption on non-alcoholic fatty liver disease. This manuscript is well-designed. In my opinion, this study is interest and could be accepted for publication in PLOS ONE after revision. I suggest that the authors take into account the following comments to revise the manuscript.

1.The abstract of the manuscript needs to be amended and the purpose of the study should be presented clearly.

2.It is suggested that the keywords be based on the "MeSH on Demand".

3.“Introduction” section: In this part, non-alcoholic fatty liver disease should be highlighted, its benefits should be mentioned more, and more references should be provided.

4.Authors should strengthen the novelty of their work in the last paragraph of introduction and conclusion section.

5.Please check all references one by one. Some references are mistake in term of formatting of this journal.

6.As the quality of the language of this manuscript is not suitable, the authors should seek the assistance of a native English speaker or a highly qualified person.

Reviewer #2: The authors aimed to assess the association between opium consumption and non-alcoholic fatty liver disease, I find the manuscript interesting, However, I have some suggestions for improvement.

Comments:

-The term "impact" is typically associated with clinical trials. I recommend changing the title to “Association Between Opium Consumption and Non-Alcoholic Fatty Liver Disease”

- It appears that the study design is a case-cohort design, as the cases derive from a cohort study while controls are selected from the baseline. Additionally, to assess the association between opium and non-alcoholic fatty liver disease, a logistic regression analysis should be conducted. The results must be rewritten based on this new analysis, and if the findings change, the discussion will need to be edited accordingly.

-there are 170 individuals newly diagnosed with NAFLD based on sonography. Therefore, the sample size in the case group should be stated as 170. A low sample size is a significant cause of low statistical power.

- The covariate regarding daily calorie intake ( kilocalories consumed in 24 hours) should clarify how this measurement was obtained.

**Do you want your identity to be public for this peer review?** For information about this choice, including consent withdrawal, please see our Privacy Policy

Reviewer #1: No

Reviewer #2: No

---

## [Author Response · Author response to Decision Letter 1]

8 May 2025

Manuscript Title: “Association of Prior Opium Addiction with Incident Non-Alcoholic Fatty Liver Disease: A Case-Control Study”

PONE-D-24-50823

We express our gratitude to the esteemed editor for granting us this opportunity, and we sincerely appreciate the invaluable contributions of the respected reviewers who dedicated their precious time to improving our manuscript. We have acknowledged our shortcomings and errors and, within a short period, made every effort to implement all the suggestions provided by the esteemed reviewers and the editor. Their feedback has been highly valuable to us, and we have incorporated the following changes into the manuscript:

Best Regards,

Dr. Farjam

• Reviewer #1

The submitted manuscript (Submission Number: PONE-D-24-50823) describes the impact of opium consumption on non-alcoholic fatty liver disease. This manuscript is well-designed. In my opinion, this study is interest and could be accepted for publication in PLOS ONE after revision. I suggest that the authors take into account the following comments to revise the manuscript.

Response:

We sincerely thank the reviewer for the thoughtful and encouraging feedback. We are pleased that you found our manuscript well-designed and of interest. Your recognition of the potential value of our research is highly appreciated and motivates us to further refine the study. We have carefully addressed all your specific suggestions in the revised manuscript and response document, and we hope the changes meet your expectations and enhance the clarity and scientific rigor of our work.

1.The abstract of the manuscript needs to be amended and the purpose of the study should be presented clearly.

Response:

Thank you for your valuable feedback. We have revised the abstract to more clearly state the purpose of the study and emphasize the time-sequenced nature of exposure and outcome assessment. Specifically, we now clarify that the study investigates whether opium addiction occurring over a six-year period (2016–2022) is associated with the new incidence and severity of NAFLD in 2022. We have also restructured the abstract for clarity and better alignment with PLOS ONE guidelines.

The new Abstract section:

“Background: Non-alcoholic fatty liver disease (NAFLD) is a prevalent chronic liver condition with no approved pharmacological treatments. Given opium’s potential metabolic effects on lipid profiles, blood pressure, and glucose levels—factors known to influence NAFLD—we hypothesized that opium addiction might be inversely associated with NAFLD risk.

Objective: To investigate the association between opium addiction during a six-year period (2016–2022) and the subsequent incidence and severity of NAFLD in 2022 among participants of the Fasa Adult Cohort Study (FACS).

Methods: Adults aged 35–70 were selected from the FACS baseline dataset (2016) after excluding individuals with NAFLD (based on the Fatty Liver Index and regional cutoffs), obesity, cancer, chronic liver diseases, or regular alcohol use. Of 550 randomly selected participants invited for sonography in 2022, 396 attended; 170 were newly diagnosed with NAFLD. Cases and controls were matched 1:1 using SPSS based on age, sex, diabetes, and hyperlipidemia. Opium addiction was defined using DSM-5 criteria via structured interviews, while NAFLD diagnosis and grading were performed using blinded ultrasound assessment. Fisher’s exact and Fisher-Freeman-Halton tests were used for analysis. A post hoc power analysis was also conducted.

Results: The final analysis included 206 participants (103 cases, 103 controls). Opium addiction was observed in 31 NAFLD cases and 72 controls, a non-significant difference. However, the prevalence of opium addiction differed significantly across NAFLD severity grades. The post hoc statistical power was estimated at 60%.

Conclusion: Although not statistically significant, fewer opium addicts developed NAFLD than non-addicts. This inverse trend, along with significant variation across NAFLD grades, suggests a possible association that warrants further investigation. Larger studies are needed to explore this potential relationship. If confirmed, opioid-based therapies may offer dual benefits for managing chronic pain and metabolic risk in selected NAFLD populations.”

2.It is suggested that the keywords be based on the "MeSH on Demand".

Response:

Thank you for your helpful suggestion. We have revised the keywords using the NIH MeSH on Demand tool to ensure they conform to MeSH terminology. The updated keywords are now: “Keywords: Opium; Behavior, Addictive; Opium; Analgesics, Opioid; Non-alcoholic Fatty Liver Disease; Noncommunicable Diseases; Pain Management; Case-Control Studies”

3.“Introduction” section: In this part, non-alcoholic fatty liver disease should be highlighted, its benefits should be mentioned more, and more references should be provided.

Response:

Thank you for your insightful suggestion. We have revised the introduction to better highlight the clinical and public health significance of NAFLD, elaborating on its progression, systemic complications, and impact in Iran and globally. Additionally, we have enriched this section by adding several key references to strengthen the scientific foundation of the introduction. These changes aim to contextualize the importance of studying NAFLD in relation to potential risk factors such as opium use.

“ Non-alcoholic fatty liver disease (NAFLD) is the most prevalent chronic disease of liver globally (Cheemerla & Balakrishnan, 2021) characterized by excessive hepatic fat accumulation in the absence of significant alcohol consumption or other secondary causes of hepatic steatosis , with an escalating incidence rate. It is estimated that currently, 32% of adults worldwide are affected by NAFLD (Teng et al., 2023). NAFLD encompasses a spectrum of liver pathology ranging from simple steatosis to non-alcoholic steatohepatitis (NASH) (Cataldo et al., 2021), which may progress to liver fibrosis, cirrhosis (a deadly condition), or liver cancer (Chrysavgis, Giannakodimos, Diamantopoulou, & Cholongitas, 2022). In addition to liver-related morbidity and mortality, it has been demonstrated to elevate the risk of cardiovascular diseases (Tana et al., 2019). Due to its largely asymptomatic nature in early stages and its association with preventable metabolic risk factors (Nd, 2019), NAFLD is increasingly recognized as a critical target for public health interventions and policy planning. Early identification and modification of modifiable risk factors are vital to mitigating its burden. Moreover, despite significant advancements in understanding the mechanisms involved in NAFLD pathogenesis and progression, there is currently no approved pharmacological treatment for it (Petroni, Brodosi, Bugianesi, & Marchesini, 2021). The management goal focuses on multidimensional weight reduction and the reduction of cardiometabolic risk factors such as blood lipid levels, blood pressure, and blood glucose (Mantovani & Dalbeni, 2021).”

4.Authors should strengthen the novelty of their work in the last paragraph of introduction and conclusion section.

Response:

Thank you for your valuable recommendation. We have revised the final paragraphs of both the Introduction and Conclusion sections to emphasize the novelty and contribution of our work. Specifically, we highlight the lack of previous studies examining the association between opium addiction and newly developed NAFLD and position our study as the first to address this gap using a population-based, matched case-control design. These revisions aim to better communicate the significance of our findings and the rationale for future research.

“Despite the growing burden of NAFLD and its intersection with metabolic health, there is limited evidence on the potential role of unconventional exposures, such as opium addiction, in its pathogenesis. Opium use remains a culturally entrenched practice in parts of the Middle East and South Asia, including Iran, where its medical and recreational consumption is historically widespread. While previous research has examined opium in relation to cardiovascular, metabolic, and neoplastic outcomes, its longitudinal association with newly developed NAFLD has not been explored. To the best of our knowledge, this is the first study to evaluate the relationship between opium addiction and incident NAFLD diagnosed via sonography in a population-based setting. By using a rigorously selected, matched case-control design within a well-characterized cohort, our study seeks to fill a critical gap in the literature and inform future longitudinal investigations on the metabolic consequences of opiate use.”

“This study is the first to explore this association using incident NAFLD cases confirmed by sonography. Our findings underscore the need for larger, prospective studies to assess the metabolic impacts of opium use and help guide public health recommendations in regions where opiate consumption is prevalent.”

5.Please check all references one by one. Some references are mistake in term of formatting of this journal.

Response:

We appreciate the reviewer’s suggestion to carefully review the reference list. We have checked each reference for accuracy and consistency with the PLOS ONE formatting requirements. Several references had inconsistencies in punctuation, author listing, journal titles, and DOI formatting, all of which have now been corrected according to the journal's style guide.

6.As the quality of the language of this manuscript is not suitable, the authors should seek the assistance of a native English speaker or a highly qualified person.

Response:

We thank the reviewer for this constructive comment. The manuscript has been thoroughly revised by a fluent English speaker with academic experience in scientific writing. Particular attention was paid to grammar, clarity, tone, and overall readability throughout the text. We believe these revisions have significantly improved the language and presentation of the manuscript. In particular, the Methods section has undergone substantial changes in terms of grammar and fluency, making it much more understandable.

• Reviewer #2

The authors aimed to assess the association between opium consumption and non-alcoholic fatty liver disease, I find the manuscript interesting, However, I have some suggestions for improvement.

Response:

We sincerely thank the reviewer for finding our manuscript interesting and for taking the time to provide constructive suggestions. We appreciate your engagement with our study and have carefully considered your feedback. The improvements made based on your recommendations aim to clarify our objectives and strengthen the presentation of our results. We are grateful for your thoughtful review and the opportunity to improve our manuscript.

-The term "impact" is typically associated with clinical trials. I recommend changing the title to “Association Between Opium Consumption and Non-Alcoholic Fatty Liver Disease”

Response:

Thank you for this helpful suggestion. We agree that the term “impact” may imply a causal inference more appropriate for interventional studies. However, as we aimed to emphasize the temporal precedence of opium addiction before NAFLD diagnosis, we have revised the title to:

“Association of Prior Opium Addiction with Incident Non-Alcoholic Fatty Liver Disease: A Case-Control Study”

This revised title avoids causal language while still reflecting the temporality and focus of our observational design.

- It appears that the study design is a case-cohort design, as the cases derive from a cohort study while controls are selected from the baseline. Additionally, to assess the association between opium and non-alcoholic fatty liver disease, a logistic regression analysis should be conducted. The results must be rewritten based on this new analysis, and if the findings change, the discussion will need to be edited accordingly.

Response:

We sincerely thank the reviewer for this constructive comment. We respectfully clarify that our study design is not a case-cohort study. It is a matched case-control study, in which both cases (incident NAFLD in 2022) and controls (NAFLD-free in 2022) were randomly selected from participants in the FACS cohort who were free of NAFLD in 2016. At follow-up in 2022, NAFLD status was assessed via ultrasonography, and opium use over the previous six years (2016–2022) was obtained via structured interviews at the same time. Therefore, both outcome and exposure data were collected in 2022, while ensuring the outcome represented new incidence.

To reduce confounding, we matched cases and controls by age, sex, diabetes status, and hyperlipidemia. Given this design, we chose to apply Fisher’s exact testing to compare the prevalence of opium use between groups, as a clear and interpretable way to explore associations without introducing model-based assumptions. While logistic regression could certainly be employed, it would require breaking the matching and reintroducing variability that we purposefully minimized. We acknowledge the reviewer’s suggestion regarding logistic regression; however, given the matched design and relatively small sample of individuals, we chose to apply fisher’s exact tests for group comparisons. This approach allowed us to directly compare the prevalence of opium use between matched cases and controls. Although logistic regression could be applied, our analytic strategy was designed to preserve interpretability and minimize overfitting given the sample size and the number of matched variables.

While we considered alternative observational designs, we ultimately selected a matched case-control design due to several key factors. Although participants were drawn from a larger cohort (FACS), prospective data on opium use were not available, and accurate exposure assessment required retrospective, in-depth interviews conducted in 2022. Given the sensitive nature of opium use and the potential for underreporting, collecting detailed exposure data at a single time point allowed for a more standardized and controlled approach. Moreover, the case-control design enabled us to efficiently control for major confounders such as age, sex, diabetes, and hyperlipidemia through matching, which would have been more difficult to achieve reliably in an unmatched cohort or cross-sectional design without requiring a substantially larger sample. Our approach also helped reduce the risk of overfitting or model instability, which could have arisen with multivariable logistic regression given the modest sample size. While we acknowledge that the statistical power of the final matched sample was limited and that the findings must be interpreted cautiously, the case-control design remains methodologically sound and efficient for evaluating the association between retrospective exposures and disease outcomes, particularly when longitudinal exposure data are unavailable.

Moreover, we note that performing logistic regression on the unmatched full sample does not alter the conclusion, the results remain statistically non-significant. However, we opted to preserve the matched design in order to highlight the need for larger, well-powered studies rather than shift the focus with a post hoc analytical change that would not influence the scientific inference. We have now clarified our rationale in the revised Methods section.

“We chose this approach to emphasize direct group comparisons after strict matching, rather than multivariable regression modeling, in order to preserve transparency in the effect of opium exposure between groups. Although regression models could be applied, our design and analytical strategy were tailored to highlight differences in exposure prevalence in a matched observational framework.”

-there are 170 individuals newly diagnosed with NAFLD based on sonography. Therefore, the sample size in the case group should be stated as 170. A low sample size is a significant cause of low statistical power.

Response:

Thank you for this important observation. While 170 individuals were indeed newly diagnosed with NAFLD based on sonography in 2022, we performed one-to-one matching with controls based on key confounding variables, a

---

## [Decision Letter · Decision Letter 1]

Association of prior opium addiction with incident non-alcoholic fatty liver disease: a case-control study

PONE-D-24-50823R1

Dear Dr. Farjam,

We’re pleased to inform you that your manuscript has been judged scientifically suitable for publication and will be formally accepted for publication once it meets all outstanding technical requirements.

Kind regards,

Borhan Mansouri

Academic Editor

PLOS ONE

Additional Editor Comments (optional):

Reviewers' comments:

Reviewer's Responses to Questions

**Comments to the Author**

Reviewer #1: All comments have been addressed

Reviewer #2: All comments have been addressed

2. Is the manuscript technically sound, and do the data support the conclusions?

Reviewer #1: Yes

Reviewer #2: Yes

3. Has the statistical analysis been performed appropriately and rigorously?

Reviewer #1: Yes

Reviewer #2: Yes

4. Have the authors made all data underlying the findings in their manuscript fully available?

Reviewer #1: Yes

Reviewer #2: Yes

5. Is the manuscript presented in an intelligible fashion and written in standard English?

Reviewer #1: Yes

Reviewer #2: Yes

Reviewer #1: The authors have responded well to all the comments and opinions of the reviewers. In my opinion, the manuscript is acceptable in its current form.

Reviewer #2: Dear Authors,

All my comments have been carefully addressed. I believe this revised version of the manuscript is suitable for publication.

**Do you want your identity to be public for this peer review?** For information about this choice, including consent withdrawal, please see our Privacy Policy

Reviewer #1: No

Reviewer #2: No

---

## [Editor Report · Acceptance letter]

PONE-D-24-50823R1

PLOS ONE

Dear Dr. Farjam,

I'm pleased to inform you that your manuscript has been deemed suitable for publication in PLOS ONE. Congratulations! Your manuscript is now being handed over to our production team.

Kind regards,

on behalf of

Dr. Borhan Mansouri

Academic Editor

PLOS ONE